# Perinatal intimate partner violence and breastfeeding practices: A systematic review and meta-analysis protocol

Zelalem Nigussie Azene[1,2]*, Catherine MacPhail[1], Lisa Gaye Smithers[1,3]

1 School of Health and Society, University of Wollongong, Wollongong, New South Wales, Australia,
2 Department of Women's and Family Health, School of Midwifery, College of Medicine and Health Sciences, University of Gondar, Gondar, Ethiopia, 3 School of Public Health, University of Adelaide, Adelaide, South Australia, Australia

* zna772@uowmail.edu.au

## Abstract

### Background

Intimate partner violence increases the risk of detrimental health, behaviors and psychological issues in mothers, affecting infant nutrition and development. However, the potential effects of maternal exposure to intimate partner violence on breastfeeding practices are understudied, and the results of individual studies are inconsistent and conflicting. The aims of this systematic review and meta-analysis are therefore to 1) estimate the prevalence of perinatal intimate partner violence and, 2) examine the relationship between perinatal intimate partner violence and breastfeeding outcomes.

### Methods and analysis

This systematic review and meta-analysis will investigate the association between perinatal intimate partner violence and breastfeeding outcomes, including early initiation within 1 hour after giving birth, exclusive breastfeeding under six months, and continued breastfeeding at two years or beyond. Comprehensive searches will be conducted in PsycInfo, Scopus, Web of Science, Medline, Cochrane, JBI EBP, CINAHL, Informit, and PubMed electronic databases. Data extraction will be performed independently by two reviewers, with discrepancies resolved by a third reviewer. Statistical analysis will be conducted using STATA/SE version 17, employing random-effects models to calculate pooled effect sizes and assess heterogeneity with $I^2$ and Chi-square tests. Subgroup analyses and meta-regression will explore potential sources of heterogeneity.

### Discussion and conclusion

Evidence suggests that intimate partner violence is linked to poor breastfeeding outcomes. This systematic review and meta-analysis will update, compile, and critically review the evidence of the role of intimate partner violence on breastfeeding outcomes. This systematic review and meta-analysis will also inform effective strategies and interventions to support breastfeeding among IPV-affected women, thereby enhancing maternal and child health.

**Data Availability Statement:** No datasets were generated or analysed during the current study. All

relevant data from this study will be made available upon study completion.

**Funding:** The author(s) received no specific funding for this work.

**Competing interests:** The authors have declared that no competing interests exist.

## Ethics and dissemination

As this review and meta-analysis involves secondary analysis of existing data, ethical approval is not required. Findings will be disseminated through peer-reviewed publications and scientific conferences, aiming to inform strategies to support breastfeeding among women affected by intimate partner violence.

## Study registration

This protocol is registered with the International Prospective Register of Systematic Reviews (PROSPERO), registration number CRD42024555048.

## Introduction

Intimate Partner Violence (IPV), the most common form of violence against women, is an important global public health concern, posing a substantial risk to the health and well-being of women [1]. It is defined as any behavior within a current or former male intimate relationship that causes physical, sexual, psychological harm, including physical aggression, sexual coercion, psychological abuse, and controlling behaviors [2, 3]. Globally, nearly one-third (30%) of women who have been in a relationship report experiencing some form of physical and/or sexual violence by their intimate partner at some point in their lifetime [1], with an increased risk of exposure during and after pregnancy [4].

Pregnancy heightens vulnerability to IPV due to changes in physical, emotional, social, and economic demands and needs. Consequently, certain risk factors may become more significant during this time, potentially causing or exacerbating violence [5]. Studies have shown that these risk factors extend beyond the pregnancy period, encompassing the timeframe from 12 months preconception to 12 months after childbirth or postpartum, and represent a period of particular vulnerability to violence [6–10]. According to the latest World Health Organization (WHO) estimate, globally, over a quarter (27%) of women aged 15–49 years who have ever been married or partnered have experienced physical or sexual violence by an intimate partner at least once in their lifetime [3]. An analysis of demographic and health surveys and the International violence against women survey found that IPV prevalence rates during pregnancy range from 2% in Australia, Denmark, Cambodia, and the Philippines to 13.5% in Uganda, with the majority ranging between 4% and 9% [4]. However, this is likely to be underreported, as producing accurate relative prevalence estimates for many conditions in women's health and healthcare, in general, remains challenging due to widespread underreporting [11]. This is particularly true for IPV, which is notoriously underreported; true cases are correctly diagnosed only about 25% of the time, with this probability varying across racial groups [12].

Perinatal IPV (P-IPV) has been linked to both fatal and non-fatal adverse health outcomes for women and their babies [13]. Non-fatal outcomes associated with IPV during pregnancy encompass a range of adverse pregnancy complications, including low birth weight, premature delivery, miscarriage, abortion, antepartum hemorrhage, intrauterine growth restriction, and perinatal death. Additionally, IPV can lead to negative health behaviors such as drug and alcohol use and smoking, as well as adverse psychosomatic outcomes including physical injuries, depression, anxiety, and suicidal tendencies [14–16]. Maternal homicide and suicide are the most extreme fatal outcomes of IPV during pregnancy [17–19]. Furthermore, IPV is suggested to adversely affect maternal breastfeeding practices particularly, in fragile settings, thereby

compromising infant and young child nutrition [20]. The relationship between IPV and breastfeeding practices can be explained through a theoretical framework called the 'deficit hypothesis', which suggests that mothers experiencing violence face physical, psychosocial, or emotional barriers to optimal breastfeeding. The deficit hypothesis indicated that IPV impacts breastfeeding through several mechanisms, including physical effects such as sore nipples and difficulty with milk let-down, psychological impacts like depression, anxiety, and body negativity, and a lack of social or partner support. Additionally, controlling behaviours associated with IPV can reduce maternal autonomy, limiting access to healthcare and breastfeeding support services, while IPV-related mental health disorders further hinder breastfeeding practices [21]. In fragile settings, the challenges related to IPV and breastfeeding are intensified due to several interconnected factors. Firstly, these settings often experience higher rates of IPV, which can adversely affect breastfeeding practices. Secondly, the limited healthcare infrastructure and lack of support services for IPV survivors and new mothers make it challenging to address both IPV and breastfeeding-related issues effectively. Additionally, food insecurity, which is prevalent in fragile settings, is strongly associated with IPV and can directly hinder a mother's ability to breastfeed [22]. Lastly, socioeconomic disadvantages, such as poverty, unemployment, and limited access to resources, further exacerbate the negative effects of IPV on breastfeeding outcomes [23, 24].

Breastmilk is the ideal food for infants, supplying all the energy and nutrients required during the initial months of life. It continues to meet up to half or more of a child's nutritional needs in the latter half of the first year and up to one-third of their needs during the second year [25]. Optimal breastfeeding is crucial for both infants and mothers. It helps reduce the incidence of childhood infections in infants, including diarrhea, ear infections, and pneumonia [26]. For mothers, breastfeeding helps lower the risk of postpartum hemorrhage and depression, reduces the likelihood of breast cancer, and decreases the incidence of premature mortality due to various infectious diseases [27, 28]. The WHO recommends initiating breastfeeding within the first hour after birth, exclusively breastfeeding for 6 months, and continuing breastfeeding for 2 years or more, alongside complementary feeding [29]. Despite the numerous benefits of maternal breastfeeding, a significant proportion of children globally are not breastfed according to the guidelines set by WHO and the United Nations Children's Fund (UNICEF) [30].

Globally, less than half of newborns begin breastfeeding within the first hour after birth and, only 41% of infants under six months are exclusively breastfed, falling short of the 2030 global target of 70%. While more than two-thirds of mothers continue breastfeeding for at least one year, the rate drops to 45% when the child reaches two years of age [31]. Although there is extensive evidence on factors influencing breastfeeding practices, psychosocial factors, such as maternal experience of IPV, remain poorly understood [30]. A recent analysis of the 2018 Demographic and Health Survey (DHS) in Cameroon found that emotional and sexual violence were linked to a reduced likelihood of early breastfeeding initiation, whereas physical violence showed no significant association. None of the dimensions of IPV were associated with exclusive breastfeeding, even after adjusting for infant, maternal, and household factors [32]. Further, in a recent study using data from the DHS of 51 low-income and middle-income countries, distributed by WHO region, 52.9% (27/51) were from Africa, 11.8% (6/51) from the Americas, 7.8% (4/51) from the Eastern Mediterranean, 11.8% (6/51) from Europe, 11.8% (6/51) from South-East Asia, and 3.9% (2/51) from the Western Pacific, found that mothers exposed to IPV were less likely to initiate breastfeeding early and breastfeed exclusively in the first 6 months.

Emerging literature on P-IPV and breastfeeding outcomes has shown divergent results. For example, a study conducted in three African countries found that sexual IPV increased the

likelihood of delayed breastfeeding initiation and reduced exclusive breastfeeding in Malawi and Zambia, while in Tanzania, mothers experiencing IPV were more likely to stop breastfeeding within the first year postpartum [30]. Similarly, a study in Pakistan highlighted the negative impact of IPV on breastfeeding practices, showing that women who experienced physical or emotional IPV were significantly less likely to exclusively breastfeed [33]. In contrast, a study in Colombia found no significant association between physical violence during pregnancy and breastfeeding outcomes, including exclusive breastfeeding, breastfeeding initiation, or any breastfeeding practices [34].

Given the significant public health implications of IPV and breastfeeding, a systematic review and meta-analysis are essential to establish a clear, evidence-based understanding of their relationship. The current literature on maternal IPV exposure and breastfeeding practices is sparse and conflicting, necessitating further investigation. Currently, there are two systematic reviews examining the association between IPV and breastfeeding practices reviewing studies published between 2001 and 2019 [35, 36]. The limitations of the first review included the inappropriate use of the Strengthening the Reporting of Observational Studies in Epidemiology (STROBE) guidelines as a quality assessment tool. STROBE is meant to improve the reporting of observational studies [37] rather than assessing their quality.

Furthermore, the review did not discuss confounding factors [36]. We hypothesise that confounding factors, such as socioeconomic status, maternal age, education, parity, and cultural norms, influence both the risk of perinatal IPV and breastfeeding outcomes by shaping maternal stress, access to resources, and support systems. These confounders create indirect pathways that must be adjusted for to accurately assess the causal relationship between perinatal IPV and breastfeeding practices. Neither review conducted a meta-analysis, highlighting a significant gap in the literature. Moreover, additional literature has recently emerged.

Hence, the aims of this review and meta-analysis are to synthesise existing evidence to determine the prevalence of P-IPV and elucidate its association with breastfeeding outcomes. Ultimately, it will inform effective strategies and interventions to support breastfeeding among IPV-affected women and suggest IPV prevention measures for perinatal women, thereby enhancing maternal and child health.

## Methods

### Protocol registration and reporting

This protocol is written according to the guidelines set forth by the Preferred Reporting Items for Systematic Reviews and Meta-Analyses (PRISMA) of observational studies in epidemiology guidelines [38] (S1 File) and is registered with the International Prospective Register of Systematic Reviews (PROSPERO), registration number CRD42024555048.

### Search strategy and information source

The search strategy will be developed using the PICO (population, intervention, control and outcomes) framework [39].

We will conduct an initial scoping search to identify relevant keywords and subject headings specific to each database and pilot test the search strategy in the following databases Medline, Scopus, and CINAHL to refine keywords and Boolean operators. The search strategy will finally be peer-reviewed by professional medical/academic librarians who are experts in systematic reviews to ensure comprehensiveness and accuracy in capturing relevant literature across the selected databases and sources. The search will be limited to studies conducted on humans and published in the English language. We plan to update the search during the process of writing the review manuscript to include new studies as they become available. Due to

their comprehensive coverage of health, medicine, nursing, and psychological literature, searches will be conducted in the following databases: PsycInfo, Scopus, Web of Science, Medline, Cochrane, JBI EBP, CINAHL, Informit, and PubMed.

Each database will be searched for the concept of "Intimate partner violence" and "Breastfeeding" using a combination of subject headings and keywords.

We will manually search the reference lists of the retrieved articles to identify and include any additional relevant studies. The search strategies for the respective databases will be detailed in supplementary data (S2 File).

## Eligibility criteria

**Inclusion criteria.** *Study design*. All peer-reviewed primary observational studies that specifically investigate the link between IPV and breastfeeding outcomes, such as early initiation, exclusivity, and duration among the perinatal population (1 year preconception, pregnant, 1 year post-partum) will be included. There will be no restrictions on the publication dates of the included studies.

**Exclusion criteria.**

- Non-peer reviewed articles such as; case reports and case series, editorials, letters to editors, commentaries, and opinion pieces without original research data, secondary or review articles, unless they contain primary data relevant to the study objectives.

- Studies focusing on populations other than the perinatal period (not including one year pre pregnancy, pregnant or 1 year postpartum women).

- Studies focusing on IPV experiences of men.

- Studies that focused on specific population groups, such as individuals with chronic diseases like HIV or eating disorders.

- Studies that focus solely on individuals identifying as sexual minorities (e.g., transgender, homosexual individuals).

*Population*. We will include women who have experienced IPV and within the perinatal period who meet the following criteria:

1. Men as perpetrators of violence against women.

2. Women who are in the postpartum period, with no contraindications to breastfeeding for either the mother or the baby.

3. Women who are both victims of IPV and perpetrators of violence towards men.

   We will exclude the following women:

1. Mothers of multiple births (women who have delivered twins, triplets, or more).

2. Women with absolute contraindications for breastfeeding (e.g., galactosemia, maple syrup urine disease, phenylketonuria, or other metabolic disorders in the infant).

3. Mothers experiencing circumstances like stillbirth, neonatal death, or other situations where breastfeeding is not possible.

4. Who are unable to breastfeed, for example, due to having undergone a mastectomy or with severe breast infections that significantly impede breastfeeding.

5. Those who have eating disorders or chronic illnesses, such as HIV, and are taking specific medications that are harmful to the infant.

6. Who have substance abuse issues, such as alcohol or drug use.

*Concept*. **Intimate partner violence** is defined as any behavior within a current or former intimate relationship with a male that causes physical, sexual, or psychological harm, including acts of physical aggression, sexual coercion, psychological abuse, economic abuse (restricting access to financial resources, bank accounts or credit cards, education, employment and health care), controlling behaviour (restricting the victim's contact with family, friends, and social networks, constantly checking on the victim's whereabouts, communications, and activities, manipulation, regulating daily life) [2, 3]. Physical violence includes intentional acts of physical force or aggression such as pushing, slapping, throwing, hair pulling, punching, kicking, or burning, use of a weapon, perpetrated with the potential to cause harm, injury, disability or death. Emotional/psychological violence is defined as acts or threats of acts, such as shouting, controlling, intimidating, humiliating, and threatening the victim. Sexual violence is defined as the use of force, coercion, or psychological intimidation to force a woman to engage in a sex act against her will whether or not it is completed.

Since IPV measurements or screening tools typically evaluate violence over the past year and lifetime [40], studies on P-IPV vary in the timeframes they cover. To gather more comprehensive data demonstrating the risks of P-IPV, we define P-IPV in this paper as an abuse experience occurring 12 months before conception, throughout pregnancy, and up to 12 months after childbirth [41, 42].

The primary outcomes of this review and meta-analysis are the first three core Infant and Young Child Feeding (IYCF) indicators established by the WHO: 1) the early initiation of breastfeeding, 2) exclusive breastfeeding under 6 months, and 3) continued breastfeeding at least 2 years or beyond [43].

Early initiation of breastfeeding will be measured as the proportion of children born in the last 24 months who were put to the breast within one hour of birth. Exclusive breastfeeding will be measured as the proportion of infants 0–5 months of age who are fed exclusively with breast milk during the previous day and given no other foods or beverages except medication. Continued breastfeeding will be measured as the proportion of women who continue breastfeeding for up to 2 years or beyond.

*Context*. Studies conducted in any geographical location, with no restrictions on region or country, and including people of all ethnicities and races, will be included.

*Screening*. The retrieved studies from the search will be uploaded to Covidence (Veritas Health Information, Melbourne, Australia), an online software where duplicates will be identified and removed and the titles and abstracts, and full-text review will be independently screened by two reviewers (ZNA and CM) [44].

If any inclusion criteria are ambiguous based on the title or abstract, the study will be selected for full-text review. The full-text screening will be performed independently by two reviewers (ZNA and CM) using the same criteria. Any discrepancies during the screening process will be resolved by a third independent reviewer (LGS). The process of study inclusion will be illustrated using a flow diagram adapted from the PRISMA guidelines [45].

*Data extraction and management*. The data extraction plan for this systematic review and meta-analysis aims to systematically gather and accurately document relevant information from the included studies to investigate the impact of P-IPV on breastfeeding outcomes. Data extraction will be conducted using Covidence. Covidence provides customisable data extraction templates that ensure consistency and comprehensiveness in capturing relevant information from the included studies. This platform also facilitates collaboration among team

members, allowing multiple reviewers to simultaneously work on data extraction while tracking changes and resolving discrepancies. The extracted data will be organised, filtered, and exported through Covidence for further analysis, ensuring a structured and consistent review process. Data extraction will be performed independently by two reviewers (ZNA and CM). Key information such as author(s), author affiliation, publication year, country of study, study design, funding sources, conflict of interest, ethics approval, sample size, participant demographics, IPV exposure details, adjustment variables, and breastfeeding outcomes will be recorded. We will contact authors for any data that is missing from publications that then allows us to include their study in our meta-analyses. We also plan to expand the data extraction forms to include measurement validation, ensuring that we capture reliable and standardised measurements from the included studies. Any discrepancies between reviewers will be resolved through discussion or decided by a third reviewer (LGS).

*Data analysis*. The extracted data will be analysed using STATA/ SETM Version 17. The analysis will begin with a descriptive summary of the included studies, detailing their key characteristics such as study design, population demographics, and outcomes. The pooled prevalence of IPV during the perinatal period will be calculated using a random-effects model.

We will assess heterogeneity among studies using the $I^2$ statistic and Chi-square test. To examine the association between P-IPV and breastfeeding outcomes, we will conduct meta-analyses to calculate pooled effect sizes. Given the anticipated variability among studies in terms of populations, settings, and methodologies, we will employ random-effects models.

To explore heterogeneity among the estimates of the primary studies, we plan to perform a subgroup analysis based on the study setting (region/s of the country where the study will be conducted e.g. Europe, Sub-Saharan Africa), high-income versus low- and middle-income countries (as defined by the UN), or specific cultural contexts where the studies were conducted), years of publication (categorised into time periods, such as pre-2010 versus post-2010, to identify any temporal trends in IPV prevalence or breastfeeding practices), sample size, and study population characteristics (include factors such as socioeconomic status, urban versus rural settings, and baseline health conditions, as reported in the included studies). Depending on the suitability of the data extracted, a meta-regression may also be performed to explore the potential sources of heterogeneity. Sensitivity analyses will ensure the robustness of our findings, and small study bias will be evaluated using funnel plots and Egger's test. Asymmetric funnel plots will suggest potential publication bias. We will consider implementing Robust Variance Estimation (RVE) to handle multiple effect sizes from the same study. To combine effect measures, we will transform all effect sizes into a common metric, such as standardised mean difference, ensuring they estimate the same treatment effect and account for design-specific sampling variance. For continuous outcomes, we will use standardised measures like effect size or standardised response mean and assess the correlation and responsiveness of different measures before pooling. In handling zero events and different follow-up periods, we will apply continuity correction or use alternative methods like Peto odds ratio for zero events. For studies with different follow-up periods, we will obtain individual participant data for time-to-event analysis when possible. We will define separate analyses for short-term, medium-term, and long-term follow-up, or select a single, clinically important time point for analysis across studies.

The plan for the treatment of adjusted vs. unadjusted estimates is as follows: For balanced covariates, both adjusted and unadjusted estimates should yield similar results. Adjusted analyses will be preferred when covariates are strong predictors or when there is an unbalanced distribution of important covariates. Caution will be taken when combining results from trials using different adjustment methods to avoid introducing heterogeneity. We plan to handle missing data by contacting study authors for unpublished data, using robust analysis methods

and imputation techniques, and conducting sensitivity analyses; address publication bias through comprehensive literature searches, funnel plots, statistical tests, and the trim and fill method.

Depending on the data in the included studies, we will use odds ratios (OR) or risk ratios (RR), along with 95% confidence intervals (CI), to determine the association between P-IPV and breastfeeding outcomes. The results will be reported following the PRISMA guidelines.

*Risk of bias (Quality) assessment.* To ensure the rigor and reliability of this systematic review and meta-analysis, we will assess the risk of bias and the quality of the included studies. The risk of bias will be assessed using the Risk Of Bias In Non-randomized Studies—of Exposures (ROBINS-E) tool. ROBINS-E is a recent tool designed for systematic reviews to evaluate bias in observational studies. It includes seven key domains: confounding, participant selection, exposure classification, deviations from intended exposures, missing data, outcome measurement, and selection of reported results. Each domain is assessed through a set of signalling questions to gather crucial information about the study and its analysis [46]. If randomised controlled trials (RCTs) are included, we will use the revised RoB 2 tool to assess the risk of bias. RoB 2 provides a framework for evaluating bias in a single estimate of intervention effects reported from RCTs [47].

We will conduct a thoughtful evaluation of each study's risk of bias categorising them as low, moderate, and high. The Grading of Recommendations Assessment, Development, and Evaluation (GRADE) approach will be used for grading the overall quality of the body of evidence [48].

Two reviewers (ZNA and CM) will independently conduct the quality assessment and compare their results. Throughout the process, discrepancies between reviewers will be resolved through discussion and consensus. If consensus cannot be reached, a third reviewer (LGS) will be consulted to make a final decision.

*Ethics and dissemination.* Ethical approval is not required for this systematic review and meta-analysis, as it involves secondary analysis of published studies that are already available in scientific databases. The results of this review and meta-analysis will be submitted for peer-reviewed publication and presented at relevant scientific meetings, national and international conferences, and seminars.

## Discussion

According to the WHO and United Nations Children's Fund Global Breastfeeding Collective Scorecard (2018), less than half of newborns initiated breastfeeding within one hour of birth, and 41% were breastfed for up to six months [49]. The target set by the collective, which is led by UNICEF and WHO is to reach a global rate of 70% for early initiation of breastfeeding, 70% for exclusive breastfeeding, and 80% for continued breastfeeding to 2 years or more by 2030. To achieve such goals, interventions, and policies at various levels are needed to promote and support breastfeeding [50]. Effective strategies include providing education and breastfeeding support before, during, and after delivery, involving not only mothers but also fathers, other family members, the health service/birthing attendant and, the broader community [51].

Breastfeeding can be influenced by a range of factors, including socioeconomic, cultural, biological, psychological, and obstetric service delivery factors [52–54]. It has also been speculated that breastfeeding outcomes can be influenced by IPV. However, the literature on the subject is still scarce, and the study results are conflicting. Some studies indicate an association between IPV and breastfeeding, while others have not found statistically significant associations [55–58]. However, many of these studies suffer from small sample sizes, poor study design, limited geographic and cultural diversity, lack of consideration for confounding

factors, and heterogeneity in study populations. Currently, there are two systematic reviews examining the association between IPV and breastfeeding practices reviewing studies published between 2001 and 2019 [35, 36]; this study expands and improves on these reviews. Collecting and synthesising evidence can be a step towards a better understanding of the impact of P-IPV on breastfeeding outcomes.

This review will synthesise the current body of evidence to explore and clarify the relationship between women's experience of IPV during the perinatal period and breastfeeding outcomes. This systematic review and meta-analysis will update, compile, and critically review the evidence on the role of P-IPV on breastfeeding outcomes. The use of GRADE will result in an overall statement on the quality of the body of evidence for each outcome. In turn, this systematic review can then be used to inform future strategies and interventions for IPV in the perinatal period to support breastfeeding, thereby enhancing maternal and child health. This review has a limitation in that non-English language articles may be excluded, potentially increasing the risk of bias. However, to maintain transparency, we intend to exclude non-English language articles during the eligibility assessment stage.

## Updates to study protocol

Should any modifications to the review protocol be necessary, these will be documented and included as supplementary information in the final manuscript and updated on the PROS-PERO register.

## Supporting information

**S1 File. Preferred Reporting Items for Systematic reviews and Meta-Analysis (PRISMA) checklist.**
(DOCX)

**S2 File. Search strategies for included databases.**
(DOCX)

## Acknowledgments

We would like to acknowledge the academic librarians (Grai Calvey and Alexandra Martins dos Santos) for their assistance in drafting the search strategies and conducting comprehensive searches across various databases.

## Author Contributions

**Conceptualization:** Zelalem Nigussie Azene.

**Data curation:** Zelalem Nigussie Azene.

**Formal analysis:** Zelalem Nigussie Azene.

**Methodology:** Zelalem Nigussie Azene, Catherine MacPhail, Lisa Gaye Smithers.

**Resources:** Zelalem Nigussie Azene.

**Software:** Zelalem Nigussie Azene.

**Supervision:** Zelalem Nigussie Azene, Catherine MacPhail, Lisa Gaye Smithers.

**Validation:** Zelalem Nigussie Azene, Catherine MacPhail, Lisa Gaye Smithers.

**Visualization:** Zelalem Nigussie Azene.

**Writing – original draft:** Zelalem Nigussie Azene.

**Writing – review & editing:** Zelalem Nigussie Azene, Catherine MacPhail, Lisa Gaye Smithers.

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
