## [Decision Letter · Decision Letter 0]

28 Nov 2024

PONE-D-24-32957Perinatal intimate partner violence and breastfeeding practices: a systematic review and meta-analysis protocolPLOS ONE

Dear Dr. Azene,

Thank you for submitting your manuscript to PLOS ONE. After careful consideration, we feel that it has merit but does not fully meet PLOS ONE’s publication criteria as it currently stands. Therefore, we invite you to submit a revised version of the manuscript that addresses the points raised during the review process.

We have received two referee reports.  The first almost solely focuses on the tense used in the paper; you do not need to change the tense as the future tense is appropriate for the protocol.  The second review has a number of useful comments about the overall theoretical framework as well as the empirical strategy that should be carefully addressed.  I will also add a few comments of my own based on my review of the paper.  First, the inclusion criteria do not seem clear or clearly justified: why are grey literature or preprints excluded?  This seems undesirable given what we know about publication bias.  I also found the criteria for the population unclear: what does it mean to refer to the "general population"?  What if there is an analysis that is a broader sample but includes many pregnant women and conducts analysis on this pregnant subsample; is this paper eligible and if not why not?  I also found the description of the subgroup analysis notably unclear: how will setting, years of publication, sample size and study population characteristics (what characteristics?) be used to define subgroups?   Meta-regression seems more appropriate given what you (appear) to have in mind, but regardless, the analytical dimensions and the empirical strategy should be described in more detail.

We look forward to receiving your revised manuscript.

Kind regards,

Jessica Leight, PhD

Academic Editor

PLOS ONE

Journal requirements: 1. When submitting your revision, we need you to address these additional requirements. Please ensure that your manuscript meets PLOS ONE's style requirements, including those for file naming. The PLOS ONE style templates can be found at https://journals.plos.org/plosone/s/file?id=wjVg/PLOSOne_formatting_sample_main_body.pdf and https://journals.plos.org/plosone/s/file?id=ba62/PLOSOne_formatting_sample_title_authors_affiliations.pdf. 2. We noticed you have some minor occurrence of overlapping text with the following previous publication(s), which needs to be addressed:- https://doczz.net/doc/7574113/intimate-partner-violence-and-pregnancy--an-intervention-...-https://doi.org/10.3389/fpsyt.2021.601236-https://doi.org/10.1371/journal.pmed.1002921In your revision ensure you cite all your sources (including your own works), and quote or rephrase any duplicated text outside the methods section. Further consideration is dependent on these concerns being addressed.

Reviewers' comments:

Reviewer's Responses to Questions

**Comments to the Author**

1. Does the manuscript provide a valid rationale for the proposed study, with clearly identified and justified research questions?

Reviewer #1: Yes

Reviewer #2: Partly

2. Is the protocol technically sound and planned in a manner that will lead to a meaningful outcome and allow testing the stated hypotheses?

Reviewer #1: Yes

Reviewer #2: Partly

3. Is the methodology feasible and described in sufficient detail to allow the work to be replicable?

Reviewer #1: No

Reviewer #2: Yes

4. Have the authors described where all data underlying the findings will be made available when the study is complete?

Reviewer #1: Yes

Reviewer #2: Yes

5. Is the manuscript presented in an intelligible fashion and written in standard English?

Reviewer #1: Yes

Reviewer #2: Yes

6. Review Comments to the Author

You may also provide optional suggestions and comments to authors that they might find helpful in planning their study.

Reviewer #1: The Methodology is in 'Future Tense' (the methodology appear like a proposal, not proper for an article submitted for publication). See the body of the article for my comments, and do the needful corrections!

Reviewer #2: As a reviewer, I find that this protocol is generally well-constructed. However, there are two key areas that warrant particular attention: (1) the theoretical framework or "theory of change" that explains the relationship between IPV and breastfeeding outcomes, and (2) the role of confounding factors within this framework.

For detailed report please see the attached file.

7. PLOS authors have the option to publish the peer review history of their article (what does this mean?). If published, this will include your full peer review and any attached files.

Reviewer #1: No

Reviewer #2: **Yes: **Jasleen Kaur

---

## [Author Response · Author response to Decision Letter 0]

16 Jan 2025

Authors’ response to editor and reviewers’ comments

Editor comments

1. First, the inclusion criteria do not seem clear or clearly justified: why are grey literature or preprints excluded? This seems undesirable given what we know about publication bias. 

Authors’ response:

While we acknowledge the potential value of grey literature and preprints in preventing reporting bias, we prioritised academic databases in the search strategy for our systematic review and meta-analysis. Indeed, a global leader in evidence synthesis have published empirical evidence showing trials that are difficult to locate are often of lower methodological quality (1). This is because grey literature and preprints often lack formal peer review and are poorly reported. Including such articles could introduce greater variability, increase bias risk, and complicate data appraisal due to inconsistent reporting. This raises the concern that extensive literature searches, rather than preventing bias, might introduce it by including low methodological quality. Egger et al suggests that when resources are limited, prioritising thorough quality assessments is more critical than conducting exhaustive searches or translating articles (1).

2. I also found the criteria for the population unclear: what does it mean to refer to the "general population"? What if there is an analysis that is a broader sample but includes many pregnant women and conducts analysis on this pregnant subsample; is this paper eligible and if not why not?

Author’s response: In our protocol, the term "general population" refers to studies that focus on women of reproductive age who are not specifically within the perinatal period (defined as one-year preconception, during pregnancy, and up to one year postpartum). These studies are excluded because our systematic review and meta-analysis aims to focus specifically on the unique relationship between women's experiences of IPV during the perinatal period and breastfeeding outcomes. For studies with a broader sample that includes many pregnant women and conducted analysis on this pregnant subsample, such papers would be eligible for inclusion. We have revised the description of the population on line number 258-259 to address this concern.

3. I also found the description of the subgroup analysis notably unclear: how will setting, years of publication, sample size and study population characteristics (what characteristics?) be used to define subgroups? Meta-regression seems more appropriate given what you (appear) to have in mind, but regardless, the analytical dimensions and the empirical strategy should be described in more detail.

Author’s response: We have revised the manuscript on lines 357-363 to include a more detail on subgroup analyses and introduced meta-regression. Briefly, the planned subgroup analyses aim to explore potential sources of heterogeneity in the included studies. Specifically:

• Setting will be used to define subgroups based on geographical regions (e.g. Europe, Sub-Saharan Africa), high-income versus low- and middle-income countries (as defined by the UN), or specific cultural contexts where the studies were conducted.

• Years of publication will be categorised into time periods, such as pre-2010 versus post-2010, to identify any temporal trends in IPV prevalence or breastfeeding practices. 

• Sample Size will be used to group studies into categories based on the size of their study populations, such as small studies versus large studies, to examine whether sample size influences the outcomes.

• Study Population characteristics will include factors such as socioeconomic status, urban versus rural settings, and baseline health conditions, as reported in the included studies.

While subgroup analyses will help explore potential sources of heterogeneity, we agree that meta-regression may be more appropriate for examining continuous variables or interactions between factors. Meta-regression would allow us to quantitatively assess the influence of these variables and their contribution to heterogeneity. Thank you for this suggestion; we will consider using meta-regression to assess heterogeneity among the included studies.

Reviewer 1 comments: 

1. Change the tense used in the paper to future tense

Author’s response: We respect the reviewer's feedback and acknowledge the comment. However, as noted by the Editor, the use of future tense is standard and appropriate for a protocol paper, as it reflects planned actions and methodologies. No changes were made to the manuscript in response to this comment. 

Reviewer 2 comments: 

Summary of Protocol: 

This systematic review and meta-analysis protocol aims to investigate the relationship between intimate partner violence (IPV) during the perinatal period and breastfeeding outcomes. The authors plan to examine early initiation, exclusive breastfeeding, and continued breastfeeding practices among women who have experienced IPV. The protocol outlines a comprehensive methodology including searches across nine major databases, rigorous screening and data extraction processes, and appropriate statistical analyses using STATA. The review will use ROBINS-E for risk of bias assessment and GRADE for evaluating the quality of evidence, with results reported following PRISMA guidelines.

Comments in short based on PLOS ONE guidelines -

1. Strengths of the Proposal: 

The protocol addresses an important public health issue with clear objectives and well-defined methodology. It demonstrates strong technical rigor in its statistical approach and includes appropriate tools for quality assessment. The authors have carefully considered various aspects of the review process, from search strategy to data synthesis, and have properly registered the protocol with PROSPERO, showing good research practice.

Comments and recommendations:

As a reviewer, I find that this protocol is generally well-constructed. However, there are two key areas that warrant particular attention: (1) the theoretical framework or "theory of change" that explains the relationship between IPV and breastfeeding outcomes, and (2) the role of confounding factors within this framework. 

1. Could you elaborate on the theoretical mechanisms explaining how IPV affects breastfeeding practices, particularly in fragile settings (Page 6)?

 Author’s response: The relationship between IPV and breastfeeding practices can be explained through a theoretical framework called the ‘deficit hypothesis’, which suggests that mothers experiencing violence face physical, psychosocial, or emotional barriers to optimal breastfeeding. The deficit hypothesis indicated that IPV impacts breastfeeding through several mechanisms, including physical effects such as sore nipples and difficulty with milk let-down, psychological impacts like depression, anxiety, and body negativity, and a lack of social or partner support. Additionally, controlling behaviours associated with IPV can reduce maternal autonomy, limiting access to healthcare and breastfeeding support services, while IPV-related mental health disorders further hinder breastfeeding practices (2). See the revised manuscript on lines 141-158

2. It would be helpful to understand why this relationship might be of higher concern in fragile areas. 

 Author’s response: In fragile settings, the challenges related to IPV and breastfeeding are intensified due to several interconnected factors. Firstly, these settings often experience higher rates of IPV, which can adversely affect breastfeeding practices. Secondly, the limited healthcare infrastructure and lack of support services for IPV survivors and new mothers make it challenging to address both IPV and breastfeeding-related issues effectively. Additionally, food insecurity, which is prevalent in fragile settings, is strongly associated with IPV and can directly hinder a mother's ability to breastfeed (3). Lastly, socioeconomic disadvantages, such as poverty, unemployment, and limited access to resources, further exacerbate the negative effects of IPV on breastfeeding outcomes (4, 5). See the revised manuscript on lines 150-158. 

3. Perhaps you could incorporate some of the emerging literature (post-2020) to strengthen the background?

Author’s response: We have now included three of the recent literature published after 2020 in the revised manuscript. 

4. Would it be possible to address how confounding factors not included in Backiewicz et al. (2020) might influence the IPV-breastfeeding relationship?

Author’s response: In their publication, Backiewicz et al wrote that there were 48 potential confounders identified from the 16 studies included in their review. However, Backiewicz et al did not list the confounders applied in each study, making it difficult for the reader to determine whether each individual study had made appropriate adjustments for confounding. If an important confounder was not included as an adjustment variable, then the IPV-breastfeeding association may be subject to residual confounding. Similarly, if a mediator has been inadvertently included as an adjustment variable, part of the total effect of IPV on breastfeeding will be ‘adjusted away’. We are applying modern epidemiological principles to the issue of confounding by preparing a Directed Acyclic Graph in advance of the review. This will help determine the potential for bias in each individual study. 

5. Could you explain how this review will build upon previous systematic reviews, particularly regarding confounding factors that Backiewicz et al. did not consider?

Author’s response: 

The review conducted by Backiewicz et al. has several limitations, outlined as follows:

1. While the review aimed to assess the association between IPV and breastfeeding outcomes, it included women exposed to other forms of violence (e.g., gang violence, bullying). Additionally, it included studies where IPV exposure occurred outside the perinatal period.

2. Women in intimate relationships lasting less than one month (during previous pregnancies, current pregnancy, or postpartum) were excluded without providing evidence to justify these exclusions.

3. The review did not explicitly list the confounders extracted from the included studies, making it unclear which specific confounders were excluded and how they might influence the IPV-breastfeeding relationship.

4. The Newcastle–Ottawa Scale (NOS) was used for quality assessment, which is nowadays considered a less rigorous tool. In contrast, we will use the ROBINS-E tool, a more robust and comprehensive quality assessment instrument.

5. The review did not perform a meta-analysis.

6. The scope of the review was not clearly defined. For example, the review included studies with IPV exposure occurring one year prior to pregnancy, during pregnancy, and postpartum, but it did not clarify the definition of the perinatal period. Furthermore, studies outside this timeframe were also included. In our systematic review and meta-analysis, we have clearly defined perinatal IPV as occurring one year preconception, during pregnancy, and up to one year postpartum.

So, this systematic review and meta-analysis will build upon previous reviews, particularly the work of Backiewicz et al., in several key ways. 

First, we will expand the analysis of confounding factors by identifying additional confounders not previously considered and constructing Directed Acyclic Graphs (DAGs) to systematically explore how these factors influence the association between perinatal IPV and breastfeeding outcomes. Additionally, we will explicitly extract and analyse confounders from all included studies, providing a clearer understanding of their role in this relationship.

Second, we will enhance quality assessment by employing the ROBINS-E tool, which is more rigorous and advanced compared to the NOS used in previous reviews. This will ensure a more robust evaluation of study quality. We will also conduct sensitivity analyses based on study quality to further strengthen our findings.

Third, our review will focus on the timing of IPV exposure, clearly defining perinatal IPV as occurring within one year preconception, during pregnancy, and one year postpartum. This precise timeframe will allow for a more accurate assessment of IPV's impact on breastfeeding outcomes. We will also examine the timing of IPV exposure, distinguishing between IPV experienced before, during, and after pregnancy.

Fourth, we will conduct detailed subgroup analyses to investigate how different types of IPV; such as physical, sexual, and emotional—impact specific breastfeeding outcomes (early initiation, exclusivity, continuation). Additionally, we will perform meta-regression analyses to quantify the influence of specific confounders on effect sizes and identify sources of heterogeneity across studies.

Fifth, unlike previous reviews, our study will include a meta-analysis to provide quantitative estimates of the association between P-IPV and breastfeeding outcomes.

Sixth, we will apply stricter inclusion criteria, focusing exclusively on P-IPV and excluding other forms of violence. We will also avoid arbitrary restrictions, such as requiring a minimum relationship duration, which ensures a more targeted and valid analysis.

Lastly, we will incorporate the most recent studies published since Backiewicz et al.’s review, ensuring a comprehensive and up-to-date synthesis of the current evidence.

6. It might be valuable to include hypotheses about confounding relationships and their role in the theoretical framework

 Author’s response: In our theoretical framework, we acknowledge that confounders such as maternal mental health, socioeconomic status, social support, and access to healthcare may influence both IPV and breastfeeding outcomes. These confounders will be accounted for in our analysis to isolate the direct effects of P-IPV. Hypotheses about their roles and pathways will be incorporated to strengthen the validity of our findings. 

Hypothesis: Confounding factors, such as socioeconomic status, maternal age, education, parity, and cultural norms, influence both the risk of perinatal IPV and breastfeeding outcomes by shaping maternal stress, access to resources, and support systems. These confounders create indirect pathways that must be adjusted for to accurately assess the causal relationship between perinatal IPV and breastfeeding practices. See the revised manuscript on lines 206-2011. 

Statistical and Methodological Considerations:

7. Have you considered implementing robust variance estimation for handling multiple effect sizes from the same study?

Author’s response: Yes, we will consider implementing robust variance estimation (RVE) to handle multiple effect sizes from the same study. We agree this is a valuable for addressing effect estimate multiplicity and dependency in meta-analyses (See line number 367-369 in the revised manuscript). 

8. Could you explain how this meta-analysis might overcome the data limitations that prevented Backiewicz et al. from conducting their meta-analysis?

Author’s response: Thank you for raising this important question. We re-read the Backiewicz et al paper and note the limitations they indicated. They did not mention any specific data-related issues, but made recommendations for IPD, called for higher quality studies and better adjustment for pre-defined confounding. We have prepared a DAG in advance to define a minimum set of confounders and, where possible, we will combine in meta-analyses the studies with and without that minimum set. This will give us some indication of bias in poorly-adjusted analyses. We will contact authors for any data that is missing from publications that then allows us to include their study in our meta-analyses. Also, we have included some studies published since the Backiewicz et al. review which can potentially increase the number of high-quality studies available for meta-analysis (See the revised manuscript on lines 341-343). 

. 

9. You might want to consider including plans for: 

o Combining different effect measures

o Handling zero events and different follow-up periods

o Treatment of adjusted vs. unadjusted estimates

Authors’ response: We address each of your suggestions as follows:

1. Regarding combining different effect measur

---

## [Editor Report · Decision Letter 1]

19 Jan 2025

Perinatal intimate partner violence and breastfeeding practices: a systematic review and meta-analysis protocol

PONE-D-24-32957R1

Dear Dr. Azene,

Thank you for your thorough response to the request for revision.  We’re pleased to inform you that your manuscript has been judged scientifically suitable for publication and will be formally accepted for publication once it meets all outstanding technical requirements.

Kind regards,

Jessica Leight, PhD

Academic Editor

PLOS ONE
---

## [Editor Report · Acceptance letter]

27 Jan 2025

PONE-D-24-32957R1 

PLOS ONE

Dear Dr. Azene, 

I'm pleased to inform you that your manuscript has been deemed suitable for publication in PLOS ONE. Congratulations! Your manuscript is now being handed over to our production team.

Kind regards, 

on behalf of

Dr. Jessica Leight 

Academic Editor

PLOS ONE